# Regional Nerve Block Decreases the Incidence of Postoperative Delirium in Elderly Hip Fracture

**DOI:** 10.3390/jcm10163586

**Published:** 2021-08-15

**Authors:** Eic Ju Lim, Won Uk Koh, Hyungtae Kim, Ha-Jung Kim, Hyun-Chul Shon, Ji Wan Kim

**Affiliations:** 1Department of Orthopedic Surgery, Chungbuk National University Hospital, Chungbuk National University College of Medicine, Cheongju 28644, Korea; limeicju@gmail.com (E.J.L.); hyunchuls@chungbuk.ac.kr (H.-C.S.); 2Department of Anesthesiology and Pain Medicine, Asan Medical Center, University of Ulsan College of Medicine, Seoul 05505, Korea; koh9726@naver.com (W.U.K.); ingwei2475@gmail.com (H.K.); Alexakim06@gmail.com (H.-J.K.); 3Department of Orthopedic Surgery, Asan Medical Center, University of Ulsan College of Medicine, Seoul 05505, Korea

**Keywords:** hip fracture, nerve block, delirium, pain, opioids

## Abstract

Postoperative delirium is common in elderly patients with hip fracture. Pain is a major risk factor for delirium, and regional nerve blocks (RNBs) effectively control pain in hip fractures. This study aimed to evaluate the effect of RNB on delirium after hip surgery in elderly patients. This retrospective comparative study was performed in a single institution, and the data were collected from medical records between March 2018 and April 2021. Patients aged ≥60 years who underwent proximal femoral fracture surgery were included, while those with previous psychiatric illness and cognitive impairment were excluded. Two hundred and fifty-two patients were enrolled and divided into an RNB or a control group according to RNB use. Delirium was assessed as the primary outcome and postoperative pain score, pain medication consumption, and rehabilitation assessment as the secondary outcomes. Between the RNB (*n* = 129) and control groups (*n* = 123), there was no significant difference in the baseline characteristics. The overall incidence of delirium was 21%; the rate was lower in the RNB group than in the control group (15 vs. 27%, respectively, *p* = 0.027). The average pain score at 6 h postoperatively was lower in the RNB group than in the control group (2.8 ± 1.5 vs. 3.3 ± 1.6, respectively, *p* = 0.030). There was no significant difference in the pain score at 12, 24, and 48 h postoperatively, amount of opioids consumed for 2 postoperative days, and time from injury to wheelchair ambulation. We recommend RNB as a standard procedure for elderly patients with hip fracture due to lower delirium incidence and more effective analgesia in the early postoperative period.

## 1. Introduction

Osteoporotic hip fracture is a major health problem because it is associated with high mortality, morbidity, and costs [1]. Although there are downward trends of mortality related to hip fracture, greater efforts are needed to achieve better outcomes [2]. The health status and health-related quality of life of elderly patients are seriously affected by the presence of hip fracture, and most patients cannot return to their performance status before injury [3].

Postoperative delirium is one of the most common complications in elderly patients with hip fracture and could result in cognitive impairment, short-term functional impairment, and increased mortality [4]. The known predisposing factors for delirium include advanced age, hip fracture surgery (in comparison to elective hip surgery), preoperative psychiatric illness, and preoperative cognitive impairment [5]. In addition, pain is a major risk factor for delirium; however, most elderly patients with hip fracture have a limited use of systemic opioid analgesics owing to side effects and their vulnerability in the drug metabolism process [6,7].

For this reason, comprehensive pain protocols have been suggested for elderly patients with hip fracture; these include evidence-based block use, timely repeated pain assessment, and multidisciplinary orthogeriatric care [8]. Regional nerve blocks (RNBs) have been proven to be effective in controlling pain in hip fractures, along with the advantages of few systemic effects [9]. Considering that pain control is essential for the reduction in delirium, we hypothesized that RNB use in hip fracture surgery would decrease postoperative pain and the incidence of delirium. This study aimed to evaluate the incidence of delirium according to RNB use in patients with hip fracture.

## 2. Materials and Methods

### 2.1. Study Population

This retrospective comparative study was performed in a single institution and approved by our institutional review board. The data were collected from medical records between March 2018 and April 2021. The inclusion criteria were as follows: (1) age of ≥60 years and (2) surgical treatment of proximal femoral fracture, which was defined as a femoral neck fracture, an intertrochanteric femoral fracture (AO/OTA 31) [10], and a subtrochanteric fracture (fracture extending 5 cm below the lower border of the lesser trochanter) [11]. Meanwhile, the exclusion criteria were as follows: (1) previous psychiatric illness, (2) previous cognitive impairment, (3) pathologic fracture, (4) prophylactic fixation, (5) revision of total hip replacement, (6) delayed surgery with neglected fracture, and (7) incomplete clinical data. Initially, 307 patients were included; ultimately, 252 patients were enrolled in this study. The patients who received general or spinal anesthesia followed by RNB were grouped into the RNB group. The patients who received only general or spinal anesthesia were grouped into the control group (Figure 1).

### 2.2. Procedures

General or spinal anesthesia was performed according to the patients’ overall health status, and RNB use was left to the discretion of the anesthesiologist. Under ultrasound guidance, a single shot of RNB, including fascia iliaca compartment block (FICB) [12], lumbar plexus block (LPB) [13], and pericapsular nerve group (PENG) block [14], was applied (Figure 2). For ultrasound-guided RNBs, a transportable ultrasound with 60 mm convex 2–5 MHz transducer for LPB and 25 mm linear 18–4 MHz transducer (Sonimage HS1TM, Konica Minolta Inc., Wayne, NJ, USA), and 21-gauge echoplex needle (Vygon, Ecouen, France) were used. After determining the insertion site, real-time ultrasound-guided perineural injection was conducted with the patient in supine or lateral position. Then, 30 mL of 0.3% ropivacaine was administered for LPB and FICB. Additionally, 20 mL of 0.3% ropivacaine was administered for PENG block.

The patients with femoral neck fracture underwent bipolar hemiarthroplasty or internal fixation with multiple screws, while those with intertrochanteric and subtrochanteric fractures underwent intramedullary nailing. After surgery, intravenous patient-controlled analgesia (PCA) was applied in all patients. On the day of surgery, intravenous acetaminophen 1 g once a day was administered unless the patient had a contraindication. If the weight of the patient was less than 50 kg, the dose of acetaminophen was controlled at 15 mg/kg. The day after surgery, oral pain medications included a tramadol 37.5 mg/acetaminophen 325 mg tablet given twice daily and tapentadol 100 mg given twice daily. Rescue analgesia during postoperative period was intravenous tramadol 50 mg or hydromorphone 0.5 or 1.0 mg. Periarticular injection was not performed. We encouraged wheelchair ambulation as soon as possible from the day after surgery, and tolerable weight-bearing ambulation started on 2 days postoperatively.

### 2.3. Data Collection

Demographic data were collected from the patients’ medical records, including age, sex, body mass index (BMI), age-adjusted Charlson comorbidity index (ACCI), Koval score before injury, injury mechanism, anesthesia method, fracture type, time from injury to surgery, time from admission to surgery, and type of surgery. The Charlson comorbidity index (CCI) was calculated by adding the coefficient assigned to comorbidities when injured. The ACCI was calculated by adding 1 point for each decade after the age of 40 years to the CCI value [15]. Falls from heights of 1 m or less were defined as “low-energy mechanism of injury” [16].

Delirium was recorded and defined using the confusion assessment method on any postoperative day or night of their hospital stay following surgery as the primary outcome [17]. Postoperative pain score, pain medication consumption, and rehabilitation assessment were evaluated as the secondary outcomes. The pain scores at 6, 12, 24, and 48 h postoperatively were assessed using the visual analog scale pain score. The consumption of pain medication for 2 postoperative days was examined. The amount of analgesics was calculated into milligrams of oral morphine according to the equianalgesic table [18,19,20]. A rehabilitation assessment was performed on the basis of the time (days) from surgery to wheelchair ambulation. All these parameters (pain score, incidence of delirium, pain medication consumption, and rehabilitation assessment) were included in the standardized protocol of our hospital.

The primary and secondary outcomes were compared between the RNB and control groups. A subgroup analysis for incidence of postoperative delirium between general and spinal anesthesia was performed. A subgroup analysis between FICB, LPB, and PENG block was performed to evaluate the differences between the blocks.

A multivariable analysis was performed to assess which variables were associated with incidence of delirium and clinically significant parameters were included in the model such as age, gender, CCI, RNB, and anesthesia type [21,22,23]. Nerve block-related complications, such as falls within 48 h after surgery and nerve injury, were assessed from the medical records. Postoperative medical complications, including deep vein thrombosis, pulmonary embolism, pneumonia, angina, myocardial infarction, and urinary tract infection, were evaluated.

### 2.4. Statistical Analysis

Categorical variables (e.g., sex, injury mechanism, method of anesthesia, fracture type, type of surgery, and incidence of delirium) were analyzed using the chi-square or Fisher’s exact test. Continuous variables (e.g., age, BMI, ACCI, Koval score, time from injury to surgery, pain score, amount of analgesics consumed, and time to wheelchair ambulation) were analyzed using an independent t-test or the Mann–Whitney test. The Shapiro–Wilk test was used to check if the data distribution was normal. The Kruskal–Wallis test was used when compare more than two continuous variables. A logistic regression analysis was conducted for multivariable analysis. All continuous data are described as means and standard deviations. Statistical significance was accepted for *p*-values of <0.05 using SPSS version 23.0 (IBM Corp., Armonk, NY, USA).

## 3. Results

The RNB group consisted of 129 patients, while the control group consisted of 123 patients. The baseline characteristics of the two study groups are presented in Table 1. There was no significant difference found in the average age, BMI, ACCI, Koval score before injury, injury mechanism, method of anesthesia, fracture type, time from injury to surgery, time from admission to surgery, and type of surgery. The methods of RNB are summarized in Table 2.

The overall incidence of delirium was 21%; the rate in the RNB group was lower than that in the control group (15 vs. 27%, *p* = 0.027). There were no nerve block-related complications, including falls within 48 h after surgery and nerve injury.

The average pain score at 6 h postoperatively in the RNB group was lower than that in the control group (2.8 ± 1.5 vs. 3.3 ± 1.6, respectively, *p* = 0.030) (Table 3). There was no significant difference in the pain score at 12 (RNB group: 2.8 ± 1.6 vs. control group: 2.7 ± 1.5, *p* = 0.432), 24 (RNB group: 2.5 ± 1.3 vs. control group: 2.4 ± 1.4, *p* = 0.154), and 48 (RNB group: 2.2 ± 1.1 vs. control group: 2.0 ± 1.1, *p* = 0.083) hours postoperatively. There was also no significant difference in the amount of opioids consumed for 2 postoperative days (oral morphine; RNB group: 36.2 ± 25.4 mg vs. control group: 33.5 ± 28.0 mg, *p* = 0.322) and time from injury to wheelchair ambulation (RNB group: 1.8 ± 2.4 days vs. control group: 1.7 ± 1.1 days, *p* = 0.407).

In the subgroup analysis between general and spinal anesthesia, there was no significant difference in the incidence of postoperative delirium (Table 4). In the subgroup analysis between FICB, LPB, and PENG block, there was no significant difference in the incidence of postoperative delirium, pain score, and amount of opioids consumed, but time to wheelchair ambulation presented a significant difference for PENG block compared with FICB and LPB (Table 5).

In the multivariable analysis, age and RNB were significantly associated with the incidence of postoperative delirium (Table 6). There was also no significant difference in the postoperative medical complications between them (Table 7).

## 4. Discussion

### 4.1. Incidence of Delirium

In this study, RNB reduced the incidence of delirium. It is widely accepted that pain reduction is important for the prevention of delirium [6], and we hypothesized that postoperative pain control could prevent delirium after hip fracture surgery. Our study results proved the effect of RNB on the incidence of postoperative delirium after hip fracture surgery. Herein, we excluded patients with a high risk of developing delirium, including those with psychiatric illness and cognitive impairment, which was based on a study by Mouzopoulos et al. [22]. They classified patients into intermediate or high risk for postoperative delirium, and FICB did not affect the incidence of delirium in patients with high risk after hip fracture surgery. In contrast, a significant reduction in the incidence of delirium (FICB group vs. placebo group: 2.4 vs. 16.9%) was observed among patients with an intermediate in their study.

The results of RNB and delirium in patients with hip fracture are inconsistent. RNB was reported to be associated with less postoperative analgesia, a lower incidence of delirium, and shorter inpatient stay [24,25]. In contrast, Guay et al. presented that there were no differences in the incidence of acute confusional state in their Cochrane review based on seven trials with 676 participants [26]. However, the study has a limitation of heterogeneity and lack of risk stratification. Unneby et al. reported that femoral nerve block (FNB) did not reduce the incidence of postoperative delirium in patients with hip fracture [27]. They focused on patients with dementia, and a large proportion developed delirium (FNB group vs. placebo group: 50/52 vs. 55/57) regardless of FNB use. This suggests that patients with major risk factors, such as cognitive and psychiatric disorders, are highly prone to developing delirium regardless of pain control. In the present study, we could prove the effect of RNB on the incidence of delirium by excluding high risk patients with the inclusion of a relatively large number of patients as a single study. Additionally, we performed a multivariable analysis with other known risk factors of delirium [21,22,23]. Age and RNB were presented to be associated with postoperative delirium, which means an appropriate risk stratification of present study. Therefore, we believe the RNB helps to prevent postoperative delirium after hip fracture surgery except in high risk elderly patients.

### 4.2. Effect on Pain Intensity and Opioid Consumption

Opioids are useful in reducing pain after surgery but have limitations regarding side effects and drug poisoning [28]. Elderly patients are known to be vulnerable to the side effects of opioids, with reductions in renal and hepatic blood flow [29]; in our study, one of our purposes was the reduction in opioid consumption with the effect of nerve block. Thompson et al. reported that preoperative fascia iliaca block significantly decreased postoperative opioid consumption [30]. They reduced the amount of tramadol by 43% and morphine by 98% with fascia iliaca block. However, contrary to our expectations, there was no significant difference in the amount of opioids consumed despite early pain reduction in our study. The first possible reason is that diverse types of opioids were prescribed with a retrospective feature. Although we calculated the equianalgesic dose of each opioid into milligrams of oral morphine [18,19,20], the results could be influenced by the diversity of the opioid types prescribed. The second possible reason is that all patients received intravenous PCA because it is a customary procedure desired by patients. In addition to additional opioids injected by nurses, rescue opioids were also administered through PCA, which were not included in the calculation of the quantity of opioids consumed. These reasons may have influenced the outcome for the quantity of opioids.

### 4.3. Functional Recovery

Based on the results reported by Marino et al., who demonstrated that continuous lumbar plexus block provided pain reduction during physical therapy [31], we expected that functional recovery could be encouraged with nerve block. However, there was no significant difference observed in the time to wheelchair ambulation between the two groups, which can be explained by some reasons. A consistent rehabilitation protocol was applied to the patients in both groups. In addition, Kim et al. demonstrated that the postoperative ambulatory capacity after hip fracture surgery is decided not by only a single factor but by multiple factors, including age, sex, preoperative ambulatory capacity, and combined medical diseases [32]. Since the postoperative ambulatory capacity is significantly associated with preoperative factors, it is possible that the reduction in pain itself could not affect the short-term functional recovery.

### 4.4. Subgroup Analysis According to Type of RNB and Anesthesia Method

In clinical practice, we usually perform LPB after induction of general anesthesia because it should be conducted in the lateral decubitus position. In contrast, FICB is usually performed before induction of spinal anesthesia because it could be conducted in the supine position, which could reduce pain for positioning of spinal anesthesia. Since LPB and FICB could block both anterior innervations of the hip joint and some surgical incision site, similar pain reduction and delirium prevention could be expected in both blocks (Figure 2). In our study, the subgroup analysis between RNBs did not demonstrate significant differences in the postoperative pain score and amount of opioids consumed, but time to wheelchair ambulation of PENG block presented significant differences compared to that of FICB and LPB. The possible reason for these results was that PENG block was introduced as a potential motor sparing analgesic block [14,33], which could encourage patients to ambulate early. However, investigation for PENG block is very limited, and validation to propose motor sparing and analgesic benefit is needed [33]. Additionally, with a relatively small number of patients included in each RNBs in the present study, detailed analysis according to block type requires careful interpretation.

There could be concerns regarding anesthesia method. Previous studies showed controversial results. Choi et al. reported general anesthesia was an independent predictor of immediate delirium [23]. In contrast, Patel et al. concluded that there was no evidence to suggest that anesthesia type influence postoperative delirium in their systematic review [34]. In the present study, there were no significant differences in the incidence of delirium between general and spinal anesthesia in subgroup analysis, and multivariate analysis showed no effect on postoperative delirium according to anesthesia method.

### 4.5. Procedure-Related Complications

Falls and procedure-related nerve injuries are important complications of RNB in patients with hip fracture [35,36]; in our study, no complication was observed. While wheelchair ambulation was encouraged from the day after surgery, the protocol of our institution was to involve at least three individuals (nurse, caregiver, and paramedics) in the transfer of a patient to a wheelchair. At 2 days postoperatively, the patients were permitted to bear weight as tolerable. Considering that the duration of a single shot of bupivacaine 0.5% (20 mL) is 22 (range, 15–32) hours [37], it is unlikely that a fall would occur during ambulation owing to RNB. Further, all RNBs were performed by experienced anesthesiologists under ultrasound guidance. We believe that our rehabilitation protocol and technique for RNB could ensure the safeness of RNB in patients with hip fracture.

### 4.6. Study Limitations

There were some limitations in this study. First, our study had a selection bias in relation to the retrospectively evaluated characteristics. RNB use was decided by the anesthesiologists, and there was no clear criterion. Future prospective studies with randomization are needed. Second, the types of nerve block used varied. FICB, LSPB, and PENG block were used herein, which might have confounded the results. Third, the severity of delirium was not considered. Some patients only demonstrated temporary inattention; however, other patients demonstrated irritability that needed restraint. We classified these demonstrations as delirium. Subgroup analyses according to the severity of delirium will be helpful in the evaluation of the effects of RNB. Despite these limitations, our study provided evidence that RNB can reduce the incidence of delirium, and we believe that its use can help improve the prognosis of elderly patients with hip fracture.

## 5. Conclusions

RNB reduced the occurrence of delirium in elderly patients with hip fracture and relieved acute pain after surgery without complications. Therefore, we recommend RNB as a standard procedure for elderly patients with hip fracture due to lower delirium incidence and more effective analgesia in the early postoperative period.

## Figures and Tables

**Figure 1 jcm-10-03586-f001:**
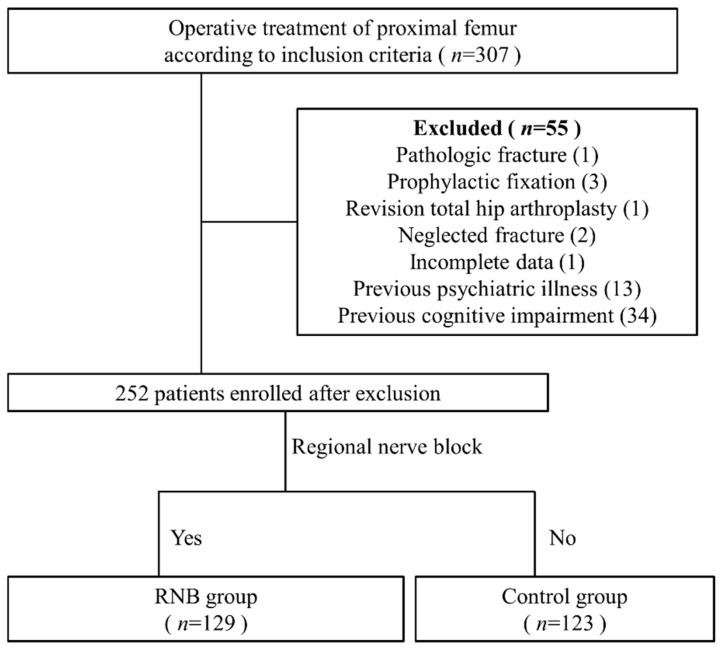
Flow diagram of patient enrollment and grouping.

**Figure 2 jcm-10-03586-f002:**
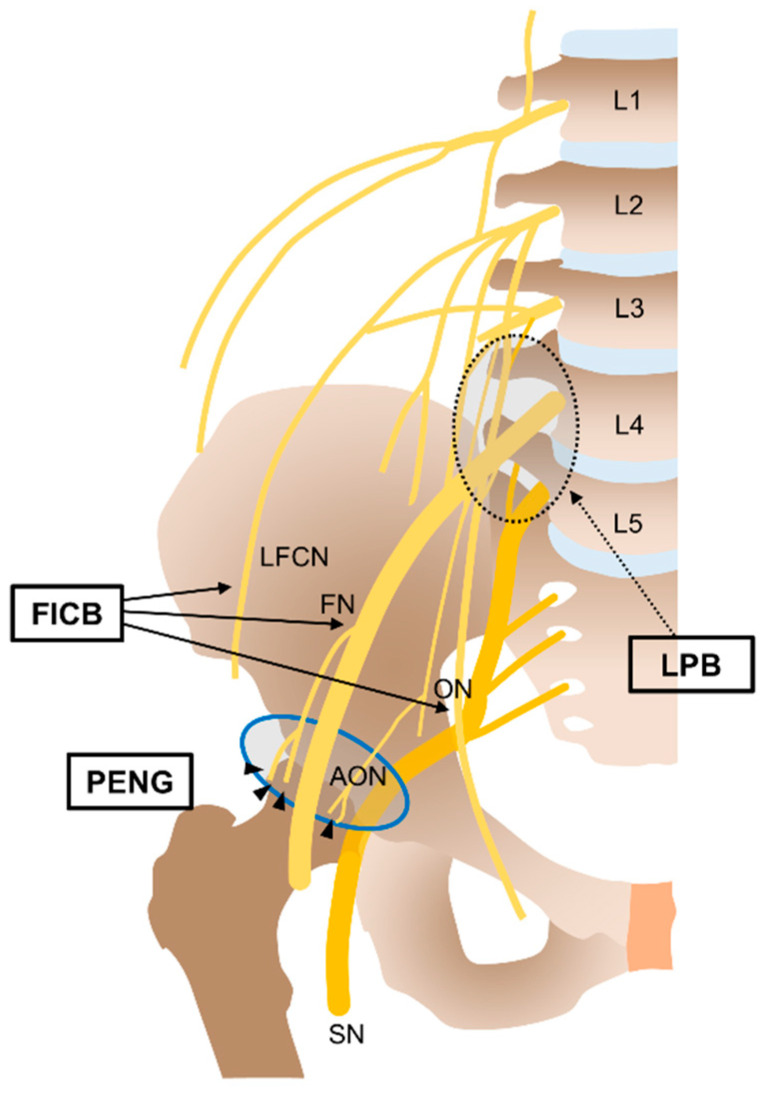
Illustration of the target nerves according to the type of regional nerve block. The fascia iliaca compartment block (FICB) targets the lateral femoral cutaneous (LFCN), femoral (FN), and obturator nerves (ON). The lumbar plexus block (LPB) targets the lumbar plexus (FN, ON, and LFCN) (dotted circle). The pericapsular nerve group (PENG) block mainly targets the articular branches (black arrowhead) of the FN and accessory obturator nerve (AON) (blue solid circle). SN: Sciatic nerve.

**Table 1 jcm-10-03586-t001:** Baseline characteristics of the RNB and control groups.

	RNB Group (*n* = 129)	Control Group (*n* = 123)	*p*-Value
Age (years)	78.1 ± 8.3	77.6 ± 8.8	0.646
Sex			
Male	32 (25%)	40 (32%)	0.175
Female	97 (75%)	83 (68%)	
BMI	22.4 ± 3.7	22.7 ± 3.8	0.560
ACCI	6.3 ± 2.0	5.9 ± 1.8	0.108
Koval score before injury *	1.9 ± 1.5	2.1 ± 1.7	0.238
Injury mechanism			
Low energy	125 (97%)	117 (95%)	0.532
High energy	4 (3%)	6 (5%)	
Anesthesia method			
General	60 (46%)	58 (47%)	1.000
Spinal	69 (54%)	65 (53%)	
Fracture type			
Femoral neck fracture	53 (41%)	58 (47%)	0.611
Pertrochanteric fracture	63 (49%)	53 (43%)	
Subtrochanteric fracture	13 (10%)	12 (10%)	
Time from injury to surgery (hour) ^†^	89.7 ± 99.4	111.0 ± 164.1	0.197
Time from admission to surgery (hour)	59.4 ± 68.7	62.6 ± 96.7	0.757
Type of surgery			
Osteosynthesis	88 (68%)	76 (62%)	0.285
Arthroplasty	41 (32%)	47 (38%)	

RNB, regional nerve block; BMI, body mass index; ACCI, age-adjusted Charlson comorbidity index; * calculated from 249 patients who had clinical records on the preoperative Koval score; ^†^ calculated from 246 patients who had clinical records on the time from injury to surgery.

**Table 2 jcm-10-03586-t002:** Type of RNB for hip fracture.

Type of RNB	
Fascia iliaca compartment block	78 (60%)
Lumbar plexus block	29 (23%)
Pericapsular nerve group block	22 (17%)

RNB, regional nerve block.

**Table 3 jcm-10-03586-t003:** Comparison of the postoperative pain score, incidence of delirium, amount of opioids consumed, and time to wheelchair ambulation.

	RNB Group (*n* = 129)	Control Group (*n* = 123)	*p*-Value
Incidence of postoperative delirium	20 (15%)	33 (27%)	0.027
Postoperative pain score			
6 h postoperatively	2.8 ± 1.5	3.3 ± 1.6	0.030
12 h postoperatively	2.8 ± 1.6	2.7 ± 1.5	0.432
24 h postoperatively	2.5 ± 1.3	2.4 ± 1.4	0.154
48 h postoperatively	2.2 ± 1.1	2.0 ± 1.1	0.083
Amount of opioids consumed (mg) *	36.2 ± 25.4	33.5 ± 28.0	0.322
Time to wheelchair ambulation (days) ^†^	1.8 ± 2.4	1.7 ± 1.1	0.407

RNB, regional nerve block; * expressed as milligrams of oral morphine by equianalgesic conversion; ^†^ calculated from 247 patients who had clinical records on the time to wheelchair ambulation.

**Table 4 jcm-10-03586-t004:** Subgroup analysis for the incidence of delirium between general and spinal anesthesia.

Incidence of Postoperative Delirium	General Anesthesia	Spinal Anesthesia	*p*-Value
RNB group	9/60 (15.0%)	11/69 (15.9%)	0.883
Control group	15/58 (25.9%)	18/65 (27.7%)	0.819
Total	24/118 (20.3%)	29/134 (21.6%)	0.800

RNB, regional nerve block.

**Table 5 jcm-10-03586-t005:** Subgroup analysis for the postoperative pain score, incidence of delirium, amount of opioids consumed, and time to wheelchair ambulation between FICB, LPB and PENG block.

	FICB (*n* = 78)	LPB (*n* = 29)	PENG Block (*n* = 22)	*p*-Value
Incidence of postoperative delirium	13 (17%)	5 (17%)	2 (9.1%)	0.786
Postoperative pain score				
6 h postoperatively	3.0 ± 1.5	2.4 ± 1.6	2.9 ± 1.5	0.215
12 h postoperatively	2.7 ± 1.6	2.7 ± 1.7	3.3 ± 1.4	0.264
24 h postoperatively	2.5 ± 1.3	2.4 ± 1.4	3.0 ± 1.0	0.161
48 h postoperatively	2.1 ± 1.1	2.3 ± 1.4	2.6 ± 0.9	0.118
Amount of opioids consumed (mg) *	35.7 ± 24.4	33.6 ± 29.2	41.6 ± 24.9	0.372
Time to wheelchair ambulation (days) ^†^	1.7 ± 1.2	2.5 ± 4.5	1.0 ± 0.2	0.020

FICB, fascia iliaca compartment block; LPB, lumbar plexus block; PENG block, pericapsular nerve group block; * expressed as milligrams of oral morphine by equianalgesic conversion; ^†^ calculated from 106 patients who had clinical records on the time to wheelchair ambulation.

**Table 6 jcm-10-03586-t006:** Bivariate and multivariable logistic regression analysis for the risk factors of postoperative delirium.

	Incidence of Postoperative Delirium			
	Bivariate Analysis		Multivariable Logistic Regression *	
	Odds Ratio (95% CI)	*p*-Value	Odds Ratio (95% CI)	*p*-Value
Age	1.060 (1.020–1.101)	0.003	1.062 (1.022–1.104)	0.002
Female sex	0.906 (0.467–1.757)	0.769		
CCI	1.091 (0.934–1.275)	0.270		
RNB	0.500 (0.269–0.932)	0.029	0.476 (0.252–0.898)	0.022
Spinal anesthesia	1.082 (0.589–1.978)	0.800		

* The results of multivariable logistic regression analysis were presented only for the variables which were remained in the final model.

**Table 7 jcm-10-03586-t007:** Postoperative complications.

	RNB Group (*n* = 129)	Control Group (*n* = 123)	*p*-Value
Deep vein thrombosis	1 (1%)	1 (1%)	>0.999
Pulmonary embolism	1 (1%)	1 (1%)	>0.999
Pneumonia	5 (4%)	4 (3%)	>0.999
Angina or myocardial infarction	1 (1%)	3 (2%)	0.360
Urinary tract infection	4 (3%)	3 (2%)	>0.999

RNB, regional nerve block.

## Data Availability

The data presented in this study are available on request from the corresponding author. The data are not publicly available due to conditions of the ethics committee of our university.

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
