# Peer review of "Regional Nerve Block Decreases the Incidence of Postoperative Delirium in Elderly Hip Fracture"

_jcm, 2021, doi:10.3390/jcm10163586_

Round 1

Reviewer 1 Report

General comments

Strenghts of the study

Very important topic. Hypothesis is correct.

Weaknesses of the study

It appears that the study is not registered in a study database and that patients have not given written consent for their medical data to be published. In this case, the study cannot be published. The ID of the ethics vote must be provided.

Nerve blocks were not standardized (which local anesthetic, which volume).

More importantly, the control group included patients with general and spinal anesthesia, who are at different risk for delirium. Thus, several studies deal with delirium incidence GA vs. SA. Moreover, patients with GA benefit from nerve block already intraoperatively, but patients with SA benefit only after spinal anesthesia has subsided.

A difference in morphine consumption was found only within the first six postoperative hours. During this period, spinal anesthesia has a significant effect on analgesic consumption.

Specific comments

Abstract

Punctuation point at the end of abstract is missing.

Introduction

Methods

2.1. study population okay

2.2. procedures

What non-opioids were given as part of the multimodal pain approach - substances, dose, all patients, as needed or fixed schedule.

If patients did not receive the same nerve block, then at least a range should be specified, e.g., 10-20ml ropivacaine 0.2%.

2.3. data collection

2.4. statistics

„The independent t-test requires that the dependent variable is approximately normally distributed within each group. You can test for this using a number of different tests, but the Shapiro-Wilks test of normality or a graphical method, such as a Q-Q Plot, are very common.“

One can also assume a normal distribution of the data if the study design is randomized.

Results

I doubt that there would be a significant difference in Pain Score 6h postoperatively using the Mann-Whitney U test (for non-normally distributed data).

Discussion

Include newer Cochrane Reviews and metaanalyses to support Your opinion. (See references)

Conclusion

Figures

Figure 2: It must be clearly indicated, that the LPNB is a double injection technique (according the reference published by de Visme et al) with different injection sides for parasacral sciatic and lumbar plexus block.

References

There are some Cochrane metaanalyses on this topic, that must be included in the reference list.

See the Reg Anesth Pain Med 2021 July Issue, that provides a perfect overview over fascial plane blocks including PENG block and Fascia iliaca compartment block.

Should the references be given at the end of the sentence (in square brackets)?

Reviewer 2 Report

Thank you for the opportunity to review this interesting paper. 

1 - In Table 1 the CCI scores are reported. The mean scores are high, 6.3 and 5.9. Bliemel et al (2016) present a mean score of 2.9, Haugan et al (2021) present a mean score of 1.17. Can you explain why your scores are so much higher?

It would be relevant to describe how the CCI scores were calculated.

Are the scores including points for age (age-adjusted)?

And, which CCI is used, the original Charlson Comorbidity Score or any updated version?

2 - Time from injury to surgery are long. Recommendations for time to surgery are 24 /48 hours. Perhaps you can include time from hospital admission to surgery to make your results more comparable to other studies.

However, time from injury is more correct to use, and often difficult to obtain, so it is valuable data. 

3 - In line 143 in your manuscript you refer to a p-value for 12 hours: 0.479 in Table 3. This p-value differ from the p-value for 12 hours in Table 3 which is 0.498.

4 - The sentence from line 174 to 176 is unclear to me. What are the numbers 2/85 and 15/89?

And, which patients were at an intermediate risk?

Do you consider all your patiens at an intermediate risk, or is this a subgroup of your patients?

5 - Line 163; the text is indicating that you miss data from 1 patient? Is this correct?

Reviewer 3 Report

Dear sirs,

I am pleased to review the paper. Overall, I think it is well conducted, but there are minor issues that prevent its imminent acceptance.

My comments are listed below,

Yours sincerely

Title:

the title is consistent with the problem actually presented, but it does not bring the main massage of the study. I would suggest considering something like this: The association of regional anesthesia decreases incidence of post op delirium in elderly hip fracture.

Abstract:

The abstract is presented in an unstructured form (according to authors' guidelines) and gives an adequate picture of the entire article.

Line 15: please consider rephrasing the sentence “this retrospective comparative study was conducted from March 2018 to April 2021 in a single…” into something like: “data were collected from medical records between March 2018 and April 2021”. Otherwise, it appears the data were collected during the whole period and it would be in contrast with the nature of a retrospective study. Please rephrase or clarify.

Introduction:

the background of the study is clear and helpful to readers unfamiliar with the subject. However, I would suggest considering adding a sentence regarding one or more studies comparing general vs spinal anesthesia for such patients. I acknowledge literature stated that there is no statistically significant difference in the incidence of delirium between the 2 techniques, but since the authors investigate the incidence of delirium in patients who underwent different anesthesia techniques, it could be helpful especially because they did not perform subgroup analysis regarding delirium in general vs spinal anesthesia. Adding such sentence/reference could also exempt the authors in performing such subgroup analysis.

(source: Patel, V., Champaneria, R., Dretzke, J., & Yeung, J. (2018). Effect of regional versus general anaesthesia on postoperative delirium in elderly patients undergoing surgery for hip fracture: a systematic review. BMJ open, 8(12), e020757.)

the purpose of the article is clearly stated at the end of the introduction

Material and Methods:

the research design is appropriate. Please use here the same advice as in line 15 in the abstract.

the criteria for selecting the sample are clearly explained and justified

the essential characteristics of the sample are adequately described

the sample size is adequate and representative

the data has been collected in a systematic and comprehensive manner

the statistical methodology is appropriate

I have no ethical concerns about this study

Chapter 2.1 study population, lines 65-66: please more clearly define the difference between the two groups. Example: patients who received general or spinal anesthesia in association with RNB -> RNB group. Patient who received general or spinal only-> control group.

Chap 2.2 procedures: regarding regional nerve blocks, please state your standard procedure regarding drug/dose/adjuvants used for such techniques.

Line 73: I have a little concern regarding the name “parasacral nerve block”. I read in line 219-220 that LPNB could block anterior+posterior innervation to the hip, so I assume that by “parasacral block” the authors mean the parasacral approach to the sciatic nerve, because the nerve to quadratus femoris and the superior gluteal nerve that provide posterior innervation to the hip emerge from the sacral plexus. (source Birnbaum, K., et al. "The sensory innervation of the hip joint-an anatomical study." Surgical and Radiologic Anatomy 19.6 (1997): 371-375.)

I would suggest to clearly state it at least the first time this nomenclature is encountered.

Results:

the analysis of the data is systematic and the results are credible.

the concurrence of symptoms/results are not likely to have arisen by chance

Since the authors collected age and gender data, it is interesting to investigate the incidence of delirium in the different subgroups (age groups, gender), as for example female gender and advanced age are prognostic factors for developing delirium. I think those additional data could add value to the paper. (source: Yang Y, Zhao X, Dong T, Yang Z, Zhang Q, Zhang Y. Risk factors for postoperative delirium following hip fracture repair in elderly patients: a systematic review and meta-analysis. Aging Clin Exp Res. 2017 Apr;29(2):115-126.)

Lines 155-158: it appears that the PENG block has been forgotten throughout data presentation, with the main focus upon FICB and LPNB. I see that there only few PENG blocks performed, but it needs to be added to the results, in the tables and in the discussion.

Discussion:

the interpretation of the results is clearly presented and adequately supported by the evidence adduced.

Conclusions:

the conclusions are logically valid and justified by the evidence adduced

Graphics:

The Tables are necessary. I already suggested to improve them adding a subgroup analysis comparing age-gender-delirium incidence.

The images instead are not relevant for the purpose of this article. If the authors prefer to provide an image, I suggest the first one only, but keeping in mind that it lacks the parasacral part (arrow or circle) of the “LPNB”.

The second image is not necessary: there is scarce view of both the US image and the anatomical landmarks, and it refers only to a single technique.

References:

the references are up-to-date and adequate. Some important previous studies have been cited, even though not all the most important ones regarding this topic.

Round 2

Reviewer 1 Report

The quality of the manuscript is significantly improved. Important concerns are eliminated.

I would recommend to structure the discussion section with subheadings: incidence of delirium, effect on pain intensity, procedure-related complications, study limitations.

Please consider to further tighten the discussion section to make it clearer for the reader (subheadings can help).

Conclusion: The authors recommend RNB as a standard procedure in elderly patient with hip fracture surgery due to lower delirium incidence and more effective analgesia in the early postoperative period.

Reviewer 3 Report

Dear authors, 

i think your paper has improved. I am glad to have been a reviewer of your work.

best regards
